# Effects of Vitamin D Supplementation During Pregnancy on Birth Size: A Systematic Review and Meta-Analysis of Randomized Controlled Trials

**DOI:** 10.3390/nu11020442

**Published:** 2019-02-20

**Authors:** Andrea Maugeri, Martina Barchitta, Isabella Blanco, Antonella Agodi

**Affiliations:** Department of Medical and Surgical Sciences and Advanced Technologies “GF Ingrassia”, University of Catania, Via S. Sofia 87, 95123 Catania, Italy; andreamaugeri88@gmail.com (A.M.); martina.barchitta@unict.it (M.B.); dott.ssa.blanco.isabella@gmail.com (I.B.)

**Keywords:** nutrition, diet, vitamin D, birthweight, birth length, head circumference, gestational age, pregnancy outcomes

## Abstract

During pregnancy, vitamin D supplementation may be a feasible strategy to help prevent low birthweight (LBW) and small for gestational age (SGA) births. However, evidence from randomized controlled trials (RCTs) is inconclusive, probably due to heterogeneity in study design and type of intervention. A systematic literature search in the PubMed-Medline, EMBASE, and Cochrane Central Register of Controlled Trials databases was carried out to evaluate the effects of oral vitamin D supplementation during pregnancy on birthweight, birth length, head circumference, LBW, and SGA. The fixed-effects or random-effects models were used to calculate mean difference (MD), risk ratio (RR), and 95% Confidence Interval (CI). On a total of 13 RCTs, maternal vitamin D supplementation had a positive effect on birthweight (12 RCTs; MD = 103.17 g, 95% CI 62.29–144.04 g), length (6 RCTs; MD = 0.22 cm, 95% CI 0.11–0.33 cm), and head circumference (6 RCTs; MD:0.19 cm, 95% CI 0.13–0.24 cm). In line with these findings, we also demonstrated that maternal vitamin D supplementation reduced the risk of LBW (3 RCTs; RR = 0.40, 95% CI 0.22–0.74) and SGA (5 RCTS; RR = 0.69, 95% CI 0.51–0.92). The present systematic review and meta-analysis confirmed the well-established effect of maternal vitamin D supplementation on birth size. However, further research is required to better define risks and benefits associated with such interventions and the potential implications for public health.

## 1. Introduction

Nutrition in women of childbearing age has a critical role on their health as well as on infant outcomes. A balanced supply of maternal nutrients, before conception, during pregnancy, and during breastfeeding, promotes optimal growth and development both in fetus and offspring [1,2]. During pregnancy, the fetus is entirely dependent on maternal sources of vitamin D, which also regulates placental function [3]. Several observational studies have shown that pregnancy is a crucial period in which vitamin D deficiency may affect mother and neonatal outcomes, thereby influencing the risk of recurrent pregnancy losses, preeclampsia, gestational diabetes, maternal infections, preterm birth, low birthweight (LBW), small for gestational age (SGA), and poor offspring health [4]. Thus, hypovitaminosis D in pregnancy requires an adequate treatment, and vitamin D supplementation represents a valid strategy for preventing and controlling vitamin D deficiency [5]. The Food and Nutrition Board at the Institute of Medicine of the National Academies suggests that a proper integration of vitamin D in pregnancy and in lactation is 15 micrograms (600 IU) per day [6]. Recently, many randomized controlled trials (RCTs) have been conducted to evaluate the benefits of vitamin D supplementation in pregnancy. Although vitamin D supplementation may increase serum 25(OH)D levels in both mothers and infants [7], it remains unclear whether vitamin D supplementation is protective for LBW, SGA or intrauterine growth restriction, and generally for long-term offspring health. To our knowledge, the meta-analysis by Thorne-Lyman et al. was the first to summarize the effect of vitamin D on birthweight and LBW incidence [8]. Pooled estimates of observational studies showed a positive relationship between vitamin D status and birthweight [8]. However, pooled analysis of interventional studies suggested, on one hand, no significant effect on mean birthweight, but on the other hand a lower risk of LBW newborns in women supplemented with vitamin D [8]. A recent systematic review by Harvey et al. [4] partially confirmed these results: out of seven studies, three trials showed that maternal vitamin D supplementation significantly increased birthweight in infants [4]. Although this evidence was also corroborated by Perez-Lopez et al. in a meta-analysis of 8 RCTs, the authors did not demonstrate the influence on the risk of LBW and SGA (3 RCTs, respectively) [5]. In contrast, four trials included in the most recent systematic review by De-Regil et al. reported that maternal vitamin D supplementation significantly reduced the risk of LBW [7]. However, no difference was reported in the mean birthweight of infants [7]. Therefore, evidence is currently inconclusive to drawn convincing assumptions on the usefulness of maternal vitamin D supplementation against LBW and SGA births. Moreover, previous mentioned reviews [4,5,7,8] showed heterogeneity in dose, duration and timing of supplementation, and study design (i.e., observational studies, quasi-RCTs, and RCTs). Herein, we provide a systematic review and a meta-analysis of data from RCTs, to evaluate the effects of oral vitamin D supplementation during pregnancy on fetal growth as indicated by neonatal anthropometric measures and incidence of LBW and SGA births. Moreover, we also performed subgroup analyses to demonstrate whether alternative formulations and regimens had different effect on birth size.

## 2. Materials and Methods

### 2.1. Literature Search

A systematic literature search in the PubMed-Medline, EMBASE, and Cochrane Central Register of Controlled Trials databases was carried out for RCTs investigating the role of oral vitamin D supplementation during pregnancy on neonatal anthropometric measures and incidence of LBW and SGA. Literature search was conducted independently by two authors (A.M. and I.B.) using the following keywords: (vitamin d OR ergocalciferol OR cholecalciferol OR calcifediol OR vitamin d supplementation OR 25-hydroxyvitamin D) AND (birth size OR birth weight OR birth length OR head circumference OR low birth weight OR SGA OR neonatal anthropometric measures). Full details of literature search terms are included in Appendix A. The databases were searched from inception to May 2017 without language restriction; abstracts and unpublished studies were not included. Moreover, the reference lists from selected articles, including relevant review papers, were searched to identify all relevant studies. The preferred reporting items for systematic reviews and meta-analysis (PRISMA) guidelines were followed [9].

### 2.2. Inclusion and Exclusion Criteria

Studies were selected only if they satisfied the following criteria: (1) RCTs, with randomization at either individual or cluster level, (2) of pregnant women of any gestational age (3) without pregnancy complications, (4) that assessed the effect of oral vitamin D supplementation, irrespective of dose, duration or time of commencement, on birthweight, birth length, head circumference, and incidence of LBW, and/or SGA. Eligible intervention groups included daily or single-intermitted vitamin D supplementation (vitamin D2 or D3), alone or in combination with calcium and/or other micronutrients. For studies with more than two intervention groups, we combined disaggregated data into subgroup category to create a single pair-wise group comparison [10]. Control groups included no treatment or placebo. Exclusion criteria were as follows: (1) systematic reviews or meta-analyses; (2) observational studies, cross-over trials, or quasi-RCTs; (3) no appropriate treatment group (pregnant women with pre-existing pregnancy complications); (4) no appropriate control group (i.e., vitamin D supplementation); (5) not available data on birth size and/or incidence of LBW and SGA.

### 2.3. Study Selection and Data Extraction

Two of the authors (A.M. and I.B.) independently assessed for inclusion all the references identified through the literature search. From all the eligible studies, two authors (A.M. and I.B.) independently extracted the following information in a standard format: first author’s last name, year of publication, country, and latitude where the study was performed, season when the study was performed, number of participants, age and information about vitamin D intervention (i.e., Formulation, regimen, method of administration, and treatment duration). Primary outcomes were birthweight (g), birth length (cm), head circumference (cm), low birthweight (LBW; <2500 g) and SGA, defined as birthweight below the 10th percentile of a reference distribution of weights specific for sex and gestational age. Other outcomes were: maternal serum 25-OHD levels at baseline and at the end of the intervention, cord 25-OHD levels, gestational age, caesarean delivery, pregnancy complications (i.e., preterm birth, gestational diabetes, pre-eclampsia), and Apgar score. Cross-checked data were entered into Review Manager software (RevMan, version 5.3, Copenhagen, Denmark)) by A.M., and checked for accuracy by M.B. During study selection and data extraction, any disagreements between A.M. and M.B. were resolved by discussion and consensus with a third Author (A.A.).

### 2.4. Risk-of-Bias and Quality Assessment

The risk of bias from each eligible RCT was evaluated using the Cochrane’s Collaboration tool for assessing risk of bias in randomized trials [10]. This tool includes the following items, which were assigned as either ‘low risk of bias’, ‘unclear risk of bias’ or ‘high risk of bias’: random sequence generation (selection bias); concealment of the allocation sequence (selection bias); blinding of participants and personnel (performance bias); blinding of outcome assessment (detection bias); incomplete outcome data (attrition bias); selective outcome reporting (reporting bias); and other biases [8]. Risk of bias was assessed by two of the authors (A.M. and M.B.) using the Review Manager software (RevMan, version 5.3), and presented as a risk-of-bias summary figure. Disagreements were resolved by consensus or discussion with a third Author (A.A.). For each outcome, two of the authors (A.M. and M.B.) independently assessed the quality of the evidence using the Grading of Recommendations Assessment, Development and Evaluation (GRADE) approach [11]. The GRADE system considers eight criteria for assessing the quality of evidence: risk of bias, inconsistency, indirectness, imprecision, publication bias, and other (large magnitude of effect, dose response, and no plausible confounding). Based on these criteria, the quality of evidence was classified as high, moderate, low, or very low. Disagreements were resolved by consensus or discussion with a third Author (A.A.).

### 2.5. Statistical Analysis

Data on dichotomous outcomes were combined and effect sizes were presented as RR with 95% confidence intervals (CIs). For continuous outcomes, we estimated mean differences (MDs) with 95% CI. Forest plots were generated to illustrate the study-specific effect sizes along with a 95% CI. The significance of pooled effect size was determined using the *Z* test, and *p* < 0.05 was considered significant. Heterogeneity across studies was assessed using the *Q*-test based on the χ2 statistic (*p* < 0.1 was considered statistically significant). To quantify heterogeneity, the I2 value was calculated and interpreted as follows: an I2 value of 0% indicates “no heterogeneity,” whereas 25% is “low,” 50% is “moderate,” and 75% is “high” heterogeneity. We considered heterogeneity as significant if *p* < 0.1 for *Q-*test based on the χ2 statistic and I² was greater than 30% [12]. To calculate pooled effect estimates, the fixed-effects model (Mantel-Haenszel method) was used in absence of significant heterogeneity, otherwise the random-effects model (Der Simonian-Laird method). In the random effect model, the between-study variance was estimated using the tau-squared (τ2) statistic [10]. The leave-one-out sensitivity analysis was performed by the omission of a single study at a time, to assess whether a particular omission could affect effect sizes and heterogeneity across studies. Sensitivity analysis by the omission of studies with daily vitamin D dose of 200 IU [13,14] was also performed to assess whether they affected the pooled effect sizes. We also performed subgroup analyses by formulation (vitamin D alone vs. vitamin D in combination with calcium and/or other micronutrients) and supplementation regimen (daily vs. single/intermitted high dose). Since vitamin D might have a dose-dependent effect on birth sizes, we performed a meta-regression using the Comprehensive Meta-analysis software (Version 2.0; Biostat Inc., Englewood, NJ, USA). Particularly, meta-regression analyses were performed on those outcomes with at least three studies evaluating daily vitamin D supplementation alone [15]. Heterogeneity in formulation, dose, duration, and timing of supplementation, did not allow us to perform meta-regression of studies with single supplementation and/or combination of supplements. In those outcomes with at least two group comparisons for subgroup, we assessed subgroup differences by the χ2 statistic, and the interaction test I² value. In those outcomes with 10 or more group comparisons, the presence of publication bias was investigated by visually assessment of funnel plot asymmetry, followed by Begg’s test and Egger’s regression asymmetry test [16,17]. Except for the *Q*-test, *p* < 0.05 was considered statistically significant, and all tests were 2-sided. All statistical analyses were performed using the Review Manager software (RevMan, version 5.3).

## 3. Results

### 3.1. Study Selection

The detailed steps of the study selection are given as a PRISMA flow diagram in Figure 1. A total of 669 abstracts were retrieved from the databases; 542 were excluded after reading titles and/or abstracts, and 127 articles were subjected to a full-text review. From these, 114 studies were excluded according to selection criteria, whereas 13 RCTs, published between 1980 and 2016, were included in the meta-analysis [13,14,18,19,20,21,22,23,24,25,26,27,28]. However, since four articles showed more eligible intervention groups [19,20,21,28], the meta-analysis reported data on 17 group comparisons between eligible intervention (*n* = 17) and control groups (*n* = 13) (Table 1).

### 3.2. Systematic Review

A total of 8 studies were from Asian countries, 4 from European countries, and 1 from USA. Accordingly, the latitude of the setting was the Northern tropic in all the included studies. To avoid confounding due to seasonal variation in sunlight exposure, 6 RCTs were carried out in different seasons [14,18,22,24,26,27]. Otherwise, the seasons varied from winter [19] to spring-summer period [13,20,25,28]; this information was not available for 2 RCTs [21,23]. Overall, sample sizes ranged between 40 and 400 pregnant women, and neonatal outcomes of 2016 newborns were reported: 1184 from mothers in the intervention groups and 832 from controls. All the eligible studies were carried out using an individual randomization. Intervention groups were characterized by vitamin D2 (*n* = 4) or D3 (*n* = 10) supplementation alone (*n* = 12) or in combination with calcium (*n* = 2) or other micronutrients (*n* = 3). Two trials did not report the vitamin D form used in 3 intervention groups [19,24]. The duration of vitamin D intervention was 6–28 weeks of gestation. Women in 11 intervention groups were supplemented with daily dose of vitamin D, whereas subjects in the remaining 6 intervention groups were supplemented with single-intermitted high dose. The daily dose ranged from 200 IU to 4000 IU; among single-intermitted interventions, the high dose varied from 35000 IU to 600000 IU. Control groups included patients who received placebo (*n* = 8) or no treatment (*n* = 5). In those studies evaluating the effect of vitamin D supplementation on 25-hydroxyvitamin D levels, the intervention significantly increased 25-OHD concentration in both mothers [13,14,18,21,22,23,25,26,27,28] and infants [21,22,25,26,27,28]. Although 3 RCTs suggested that women who received vitamin D supplementation during pregnancy had a lower risk of preterm birth than controls [13,24,27], in other trials the intervention did not affect gestational age [13,14,20,21,22,24] and preterm birth risk [14,19,22].

### 3.3. Meta-Analysis

#### 3.3.1. Birthweight

The effect of vitamin D supplementation on birthweight was assessed by 12 RCTs [13,14,18,19,20,21,22,23,24,25,26,27] and 15 group comparisons. Compared to controls, birthweight was significantly higher in the intervention groups (MD: 103.17 g, 95% CI 62.29–144.04 g; *p* < 0.001). *Q*-test and I2 statistics showed no significant heterogeneity across studies (*p* > 0.1; I2 = 7.0%). Particularly, vitamin D supplementation alone, but not in combination with other micronutrients, significantly increased birthweight (MD: 118.46 g, 95% CI 70.47–166.45 g, *p* < 0.001; MD: 62.76 g, 95% CI −15.24–140.77 g, *p* = 0.520, respectively) (Figure 2). The effect of vitamin D in combination with other micronutrients remained no significant also after the omission of studies with daily vitamin D dose of 200 IU [13,14] (MD: 49.30 g, 95% CI −43.52–142.11 g, *p* = 0.300). Particularly, meta-regression did not reveal a dose-dependent effect of vitamin D supplementation alone on birthweight (*p* = 0.773). Subgroup analysis by regimen showed that both daily and single-intermitted high dose supplementation significantly increased birthweight (MD: 74.66 g, 95% CI 18.80–130.52 g, *p* < 0.001; MD: 136.02 g, 95% CI 76.05–195.98 g, *p* < 0.001, respectively).

#### 3.3.2. Birth Length

The effect of vitamin D supplementation on birth length was assessed by 7 RCTs [13,18,19,22,23,25,26] and 8 group comparisons. The *Q*-test and I2 statistics showed significant heterogeneity across studies (*p* < 0.001; I2 = 75.2%). Based on the random effect model, birth length was significantly higher in the intervention groups than in controls (MD: 0.50 cm, 95% CI 0.08–0.92 cm; *p* = 0.020). Moreover, we performed a leave-one-out sensitivity analysis to investigate the sources of heterogeneity across studies. The sensitivity analysis found that the study by Marya et al. [23] affected the heterogeneity across studies. When this study was omitted, the between-studies heterogeneity decreased (*p* > 0.1; I2 = 0.0%) and birth length remained significantly higher in the intervention groups than in controls (MD: 0.22 cm, 95% CI 0.10–0.34 cm; *p* < 0.001). Meta-regression did not reveal a dose-dependent effect of vitamin D supplementation alone on birth length (*p* = 0.895). Subgroup analysis by formulation showed that vitamin D supplementation alone, but not in combination with other micronutrients, significantly increased birth length (MD: 0.22 cm, 95% CI 0.10–0.34 cm, *p* < 0.001; MD: 0.21 cm, 95% CI −0.49–0.92 cm, *p* = 0.700, respectively) (Figure 3). Subgroup analysis by regimen showed that both daily and single-intermitted high dose supplementation significantly increased birth length (MD: 0.20 cm, 95% CI 0.08–0.32 cm; *p* = 0.001; MD: 0.50 cm, 95% CI 0.02–0.97 cm; *p* = 0.041, respectively).

#### 3.3.3. Head Circumference

The effect of vitamin D supplementation on head circumference was assessed by 6 RCTs [13,14,18,19,22,25] and 7 group comparisons. Compared to controls, head circumference was significantly greater in the intervention groups (MD: 0.19 cm, 95% CI 0.13–0.24 cm; *p* < 0.001). *Q-*test and I2 statistics showed no significant heterogeneity across studies (*p* > 0.1; I2 = 2.0%). Particularly, vitamin D supplementation alone, but not in combination with other micronutrients, significantly increased head circumference (MD: 0.19 cm, 95% CI 0.14–0.25 cm, *p* < 0.001; MD: −0.06 cm, 95% CI −0.41–0.28 cm, *p* = 0.720, respectively; Figure 4). Meta-regression did not reveal a dose-dependent effect of vitamin D supplementation alone on head circumference (*p* = 0.746). Subgroup analysis by regimen showed that daily maternal vitamin D supplementation significantly increased head circumference (MD: 0.19 cm, 95% CI 0.14–0.24 cm; *p* < 0.001). Regarding single-intermitted intervention, Roth et al. reported no significant effect [25].

#### 3.3.4. Low Birthweight

The effect of vitamin D supplementation on incidence of LBW was assessed by 3 RCTs [14,18,28] and 4 group comparisons. Compared to control groups, the risk of LBW newborns was lower in the intervention groups (RR = 0.40, 95% CI 0.22–0.74; *p* = 0.003) (Figure 5). Interestingly, the omission of the study by Brough [14], showed that women supplemented with vitamin D alone had a lower risk of LBW than controls (RR = 0.47, 95% CI 0.23–0.97; *p* = 0.040). In contrast, Brough et al. reported no significant effect of vitamin D supplementation with micronutrients [14]. *Q-*test and I2 statistics showed no significant heterogeneity across studies (*p* > 0.1; I2 = 0%). Subgroup analysis by regimen showed that daily maternal vitamin D supplementation significantly reduced the risk of LBW (RR = 0.40, 95% CI 0.21–0.78; *p* = 0.007). Regarding single-intermitted intervention, Yu et al. reported no significant effect [28].

#### 3.3.5. Small for Gestational Age

The effect of vitamin D supplementation on incidence of SGA was assessed by 5 RCTs [14,18,22,27,28] and 6 group comparisons. Compared to control groups, the risk of SGA was lower in the intervention groups (RR = 0.69, 95% CI 0.51–0.92; *p* = 0.018) (Figure 6). Interestingly, the omission of the study by Brough [14], showed that women supplemented with vitamin D alone had a lower risk of SGA than controls (RR = 0.70, 95% CI 0.47–0.97; *p* = 0.047). In contrast, Brough et al. reported no significant effect of vitamin D supplementation with micronutrients [14]. *Q-*test and I2 statistics showed no significant heterogeneity across studies (*p* > 0.1; I2 = 16.2%). Meta-regression did not reveal a dose-dependent effect of vitamin D supplementation alone on SGA (*p* = 0.903). Subgroup analysis by regimen demonstrated that both daily and single-intermitted dose significantly reduced the risk of SGA (RR = 0.73, 95% CI 0.51–0.98; *p* = 0.042; RR = 0.58, 95% CI 0.32–0.99; *p* = 0.048).

### 3.4. Risk-of-Bias and Quality Assessment

Risk-of-bias assessment was shown in Figure 7. Overall, we identified low risk of attrition and other biases; however, unclear and/or high risk of selection, performance, detection, and reporting biases cannot be excluded. Overall, the quality of evidence varied from very low (head circumference) to moderate (birthweight, birth length, LBW, and SGA). The main reasons for downgrading the quality of evidence were the risk of bias of RCTs (i.e., high risk of bias for blinding) and the imprecision (i.e., low sample size and/or number of events which resulted in wide 95% CI). Full details of risk-of-bias and quality assessment are included in Appendix A.

### 3.5. Publication Bias

The funnel plot of the effect of vitamin D supplementation on birthweight appeared symmetric (Figure 8). Accordingly, Begg’s rank correlation method and Egger’s weighted regression method showed no sources of publication bias (*p*-values > 0.05). Regarding other outcomes, due to the limited number of studies, the extent of publication biases cannot be excluded because the power of the tests for funnel plot asymmetry was too low to identify a real asymmetry [10].

## 4. Discussion

When interpreting findings about the effect of maternal vitamin D supplementation on birth size, differences in study design and type of intervention should be considered. To reduce heterogeneity, we limited our analysis to RCTs of pregnant women without pregnancy complications, with randomization at either individual or cluster level. We excluded observational studies, cross-over trials, or quasi-RCTs. Moreover, to demonstrate whether alternative formulations and/or regimens had different effect on birth size, we performed subgroup analyses. Overall, we compared the effect of oral vitamin D (D2 or D3 form) supplementation, alone or in combination with other micronutrients, with placebo or no treatment. Most of the interventions provided daily vitamin supplementation, and to a lesser extent intermitted or single high doses. To date, the recommended dose of vitamin D supplementation, during pregnancy and lactation, is 600 IU per day [6]. This recommendation, based on outcomes related to skeletal health, was proposed as the amount of vitamin D to maintain blood levels of 25(OH)D above 50 nmol/L [29]. However, several lines of evidence argued that deficiency should be defined at thresholds of 75 nmol/L or higher [30,31,32,33,34]. The majority of RCTs included in our work evaluated the benefits of a supplementation greater than 600 IU per day. The daily doses ranged from 200 IU to 4000 IU while, among high single-intermitted interventions, the doses varied from 35000 IU to 600000 IU. During pregnancy, the tolerable upper intake level of vitamin D is now set at 4000 IU/day, although the adverse effects of higher levels are uncertain [30]. As suggested by several RCTs included in the present systematic review, interventions which reached or exceed the tolerable upper intake level did not manifest adverse effects [20,22,23,25,26,28]. However, due to inconclusive evidence, monitoring of toxicity and potential adverse effects of high intermittent dosages should be an important consideration in RCTs design [30]. 

Our findings add to the growing body of evidence about the effect of vitamin D supplementation on neonatal anthropometric measures and incidence of LBW and SGA births. Compared to previous meta-analyses [5,8], our work summarized data from 13 RCTs, published until May 2017, providing a larger number of group comparisons for each outcome of interest. We confirmed that vitamin D supplementation alone, but not in combination with other micronutrients, significantly increased birthweight, birth length, and head circumference. Compared to the most recent meta-analysis of RCTs by Perez-Lopez et al. [5], we also showed that newborns from women supplemented with vitamin D alone had a lower risk of LBW. This evidence was consistent with four trials included in the recent systematic review by De-Regil et al. [7] and pooled results by Thorne-Lyman et al. [8]. Although birthweight and length are the most used indicators for the assessment of intrauterine growth, the evaluation of weight distribution at birth is more adequate when it is adjusted for gestational age. Accordingly, the reduction of LBW is not the best goal of intervention of RCTs, because it does not distinguish between suboptimal fetal growth and shortened gestation [35]. Evidence about the effect of vitamin D supplementation on gestational age and preterm birth is controversial; De-Regil et al. reported that vitamin D supplementation alone may reduce the risk of preterm birth, while the combined supplementation with calcium increased the risk [7]. Findings from RCTs included in the present study suggested that women who received vitamin D supplementation alone [24,27] or in combination with calcium [13] had a lower risk of preterm birth. However, in other RCTs the intervention did not affect gestational age [13,14,20,21,22,24] and preterm birth incidence [14,19,22]. To our knowledge, the present study is the first demonstrating that maternal vitamin D supplementation alone significantly reduced the risk of SGA births, defined as birthweight below the 10th percentile for sex and gestational age. SGA newborns show a higher risk of neonatal and infant mortality, childhood malnutrition, neurocognitive disorders, and adulthood metabolic diseases [36]. Subgroup analysis by regimen showed that both daily and high single-intermitted dose significantly increased birthweight and length, reducing the risk of SGA births. In fact, the fat-soluble properties of vitamin D allow the single-intermitted dosage during pregnancy, which may be a feasible strategy against adverse neonatal outcomes in low income countries with poor health infrastructures [8].

Since we included studies with different interventions, we assessed the dose-dependent effect of vitamin D supplementation on birth size. For this purpose, we performed meta-regression analyses of those outcomes with at least three studies evaluating daily vitamin D supplementation alone. By contrast, heterogeneity in formulation, dose, duration, and timing of supplementation, did not allow us to perform meta-regression of studies with single supplementation and/or combination of supplements. However, our analysis did not reveal a dose-dependent effect of vitamin D on birth sizes, probably due to the limited number of studies. Moreover, very few studies investigated the effect of vitamin D supplementation with other micronutrients, and hence we were not able to understand whether the combination with other supplements might affect the efficacy of intervention. Although single and combination interventions were based on similar vitamin D doses, we cannot exclude an antagonistic effect of other micronutrients. Given these limitations, further research is needed to assess the dose-dependent effect of vitamin D alone or in combination with other micronutrients. 

The positive effect of maternal vitamin D supplementation on birth size and risk of LBW and SGA might be mediated by changes in fetal cell mass and function, skeletal mineralization, and metabolism [37]. Moreover, maternal serum vitamin D insufficiency is associated with an increased risk of preterm birth [38,39,40]. The main role of vitamin D in the human body is to maintain adequate levels of calcium and phosphate, enabling the critical processes of bone mineralization and development during fetal life [37]. In many fetal tissues, the active form of vitamin D binds to the vitamin D receptors, regulating genes responsible for the proper implantation of the placenta [41], which is important for fetal growth. Moreover, vitamin D could influence the maternal immune response to the placenta [42] and the expression of human chorionic gonadotropin and sex steroids [43]. Some experimental studies have also proposed the role of vitamin D in glucose and insulin metabolism, affecting availability of energy to the fetus [44], as well as its influence on musculoskeletal growth [45]. 

Potential weaknesses of our work include the limited number of databases searched. According to selection criteria, some data, such as conference abstracts and/or unpublished reports, were excluded. Our reluctance to include unpublished results is based on: (i) the absence of peer-review of unpublished literature; (ii) the studies that can be located may be an unrepresentative sample of all unpublished studies; (iii) unpublished studies may be of lower methodological quality than published studies [10]. To address publication biases, we visually assessed funnel plot asymmetry followed by Begg’s test and Egger’s regression asymmetry test. However, in those outcomes with less than 10 intervention groups, the extent of publication bias cannot be completely excluded. Beyond potential reporting and publication biases, we assessed an unclear risk of bias for random sequence generation, allocation concealment, selective reporting and blinding (i.e., absence of blinding should be considered when interpreting results by Hossain et al. [22]). Accordingly, the quality of evidence varied from very low to moderate. The main reasons for downgrading the quality of evidence were the high risk of bias for blinding and the low sample size, which resulted in wide 95% CI. We also recognize that including more group comparisons could have the potential of overestimating or underestimating the effect of vitamin D supplementation. To solve this issue, where possible, we performed subgroup analyses that classified group comparison into different subgroups, whenever possible. Another weakness is the potential effect of confounders. To control for factors that could contribute to the effect of vitamin D supplementation on birth size, RCTs should be based on more standardized protocols [46]: dosage for vitamin D supplementation should be chosen upon the maternal serum 25OHD levels at baseline [8], as performed by Sablok et al. [27], and such trials should take into account risk factors for vitamin D deficiency (i.e., genetic factors, latitude, lifestyles, dietary intake and seasonality) [47,48,49]. To avoid confounding due to seasonal variation in sunlight exposure, several RCTs were carried out in different seasons [14,18,22,24,26,27]. Taking into account the abovementioned limitations, further research and large multicenter RCTs, evaluating the effect of genetic, environmental, sociodemographic, and life-style factors, is needed.

## 5. Conclusions

Despite growing interest in the relationship between vitamin D supplementation and pregnancy outcomes, previous evidence about the effect on birth size remained weak. Findings from these systematic review and meta-analysis confirm vitamin D as an essential nutrient for fetal growth and development, with well-established effects on birth size. Moreover, to our knowledge, this work represents the first meta-analysis of RCTs which demonstrates a significant positive effect of maternal vitamin D supplementation on the risk of SGA births. However, further RCTs of vitamin D supplementation during pregnancy are required to better define risks and benefits associated with such interventions and the potential implication as a feasible strategy to prevent adverse pregnancy outcomes.

## Figures and Tables

**Figure 1 nutrients-11-00442-f001:**
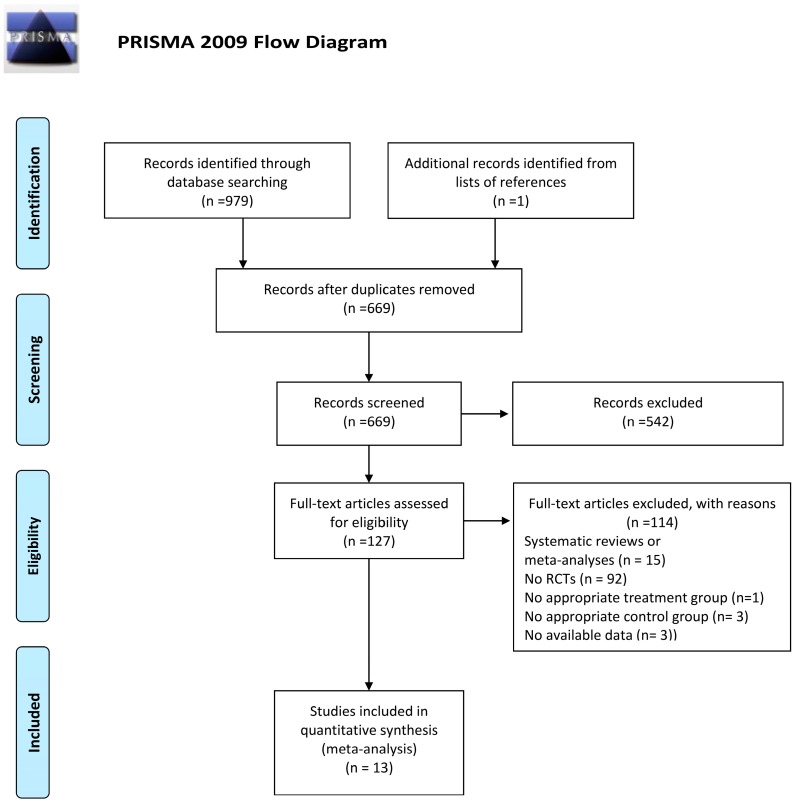
PRISMA flow diagram of study selection.

**Figure 2 nutrients-11-00442-f002:**
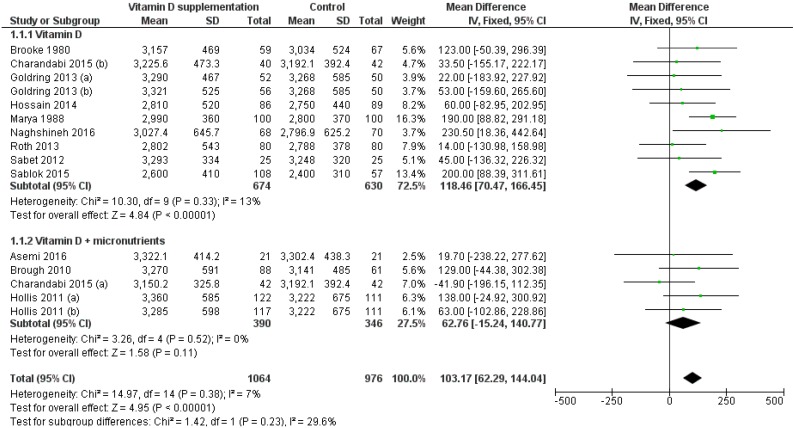
Forest plot of the effect of vitamin D intervention alone or in combination with micronutrients on birthweight (g), based on the fixed-effects model. Charandabi 2015 (**a**): Vit D + Ca (1000 IU/day); Charandabi 2015 (**b**): Vit D (1000 IU/day); Goldring 2013 (**a**): Vit D2 (800 IU/day); Goldring 2013 (**b**): Vit D3 (single dose of 200000 IU); Hollis 2011 (**a**): Vit D3 + micronutrients (1600 IU/day); Hollis 2011 (**b**): Vit D3 + micronutrients (3600 IU/day).

**Figure 3 nutrients-11-00442-f003:**
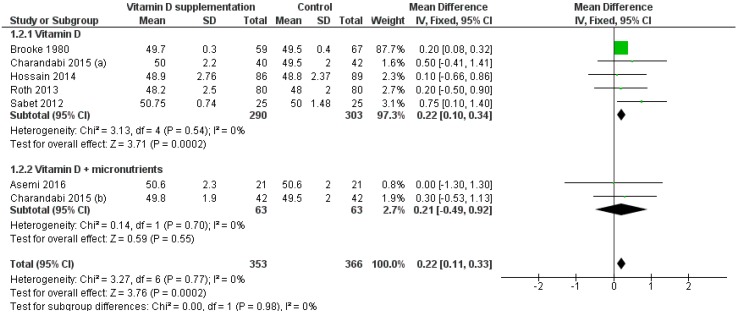
Forest plot of the effect of vitamin D intervention alone or in combination with micronutrients on birth length (cm), based on the fixed-effects model. Charandabi 2015 (**a**): Vit D + Ca (1000 IU/day); Charandabi 2015 (**b**): Vit D (1000 IU/day).

**Figure 4 nutrients-11-00442-f004:**
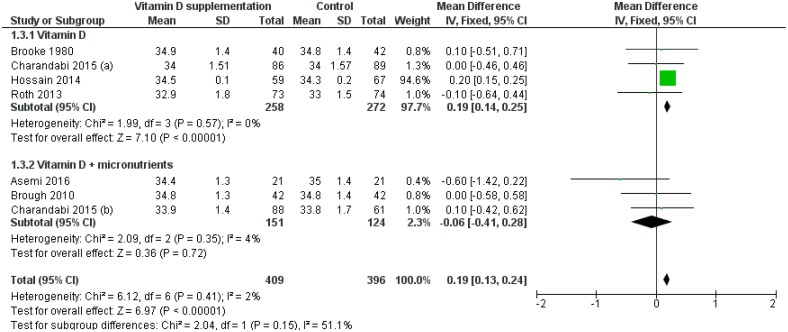
Forest plot of the effect of vitamin D intervention alone or in combination with micronutrients on head circumference (cm), based on the fixed-effects model. Charandabi 2015 (**a**): Vit D + Ca (1000 IU/day); Charandabi 2015 (**b**): Vit D (1000 IU/day).

**Figure 5 nutrients-11-00442-f005:**
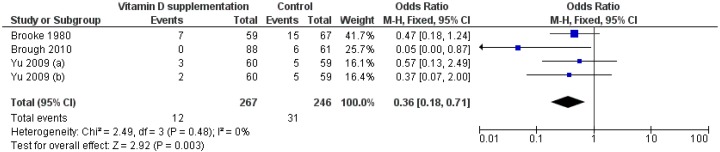
Forest plot of the effect of vitamin D intervention on the risk of low birthweight, based on the fixed-effects model. Yu 2009 (**a**): Vit D2 (800 IU/day); Yu 2009 (**b**): Vit D2 (single dose of 200000 IU).

**Figure 6 nutrients-11-00442-f006:**
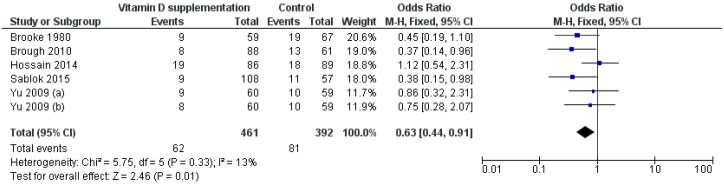
Forest plot of the effect of vitamin D intervention on the risk of small for gestational age, based on the fixed-effects model. Yu 2009 (**a**): Vit D2 (800 IU/day); Yu 2009 (**b**): Vit D2 (single dose of 200000 IU).

**Figure 7 nutrients-11-00442-f007:**
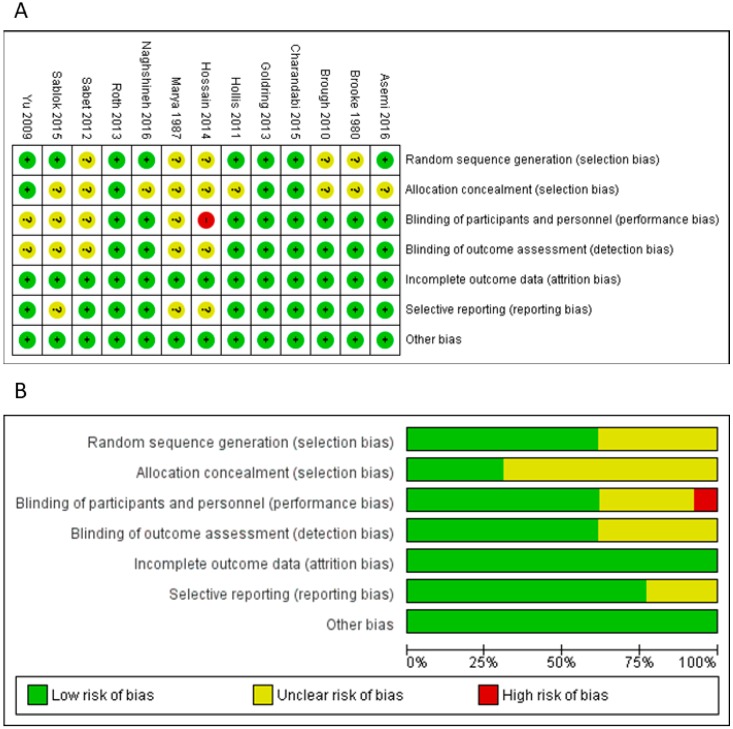
Risk-of-bias assessment of randomized controlled trials included in the meta-analysis. Risk-of-bias summary (**A**) and graph (**B**), according to the Cochrane’s Collaboration tool for assessing risk of bias in randomized trials.

**Figure 8 nutrients-11-00442-f008:**
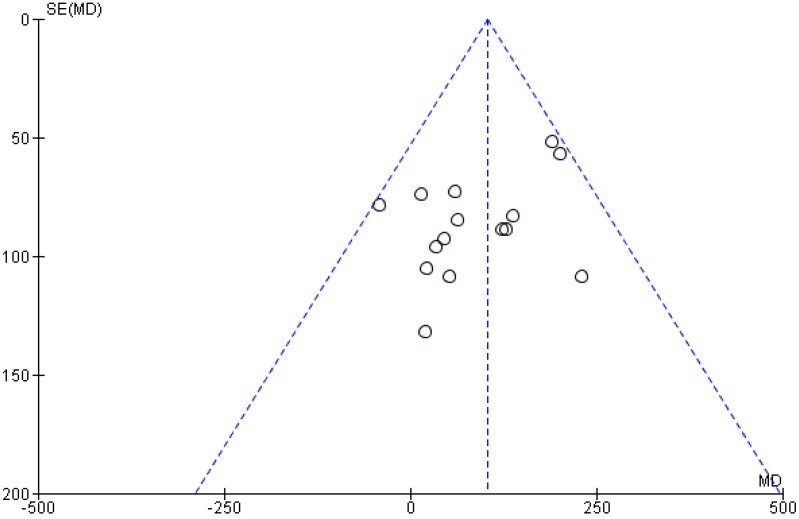
Funnel plot with estimated 95% confidence intervals for meta-analysis of the effect of vitamin D intervention on birthweight.

**Table 1 nutrients-11-00442-t001:** Characteristics of randomized controlled trials included in the meta-analysis.

First Author, Year	Country	Treatment (Vitamin D Dosage)	CONTROL GROUP	Size of Intervention/Control Groups	Treatment Duration (Week)	Outcomes
Asemi, 2016 [13]	Iran	Vit D3 + Ca (200 IU/day)	Placebo	21/21	9	Birthweight, birth length, head circumference
Brooke, 1980 [18]	UK	Vit D2 (1000 IU/day)	Placebo	59/67	8–12	Birthweight, birth length, head circumference, LBW, SGA
Brough, 2010 [14]	UK	Vit D3 + micronutrients (200 IU/day)	Placebo	88/61	NA	Birthweight, head circumference, LBW, SGA
Charandabi, 2015 [19]	Iran	Vit D + Ca (1000 IU/day)	Placebo	40/42	9	Birthweight, birth length, head circumference
		Vit D (1000 IU/day)		42/42		
Goldring, 2013 [20]	UK	Vit D2 (800 IU/day)	No intervention	56/50	12	Birthweight
		Vit D3 (single dose of 200000 IU)				
Hollis, 2011 [21]	USA	Vit D3 + micronutrients (1600 IU/day)	Placebo	122/111	24–28	Birthweight
		Vit D3 + micronutrients (3600 IU/day)		117/111		
Hossain, 2014 [22]	Pakistan	Vit D3 (4000 IU/day)	No intervention	86/89	16	Birthweight, birth length, head circumference, SGA
Marya, 1988 [23]	India	Vit D3 (two doses of 600000 IU)	No intervention	100/100	12	Birthweight, birth length
Naghshineh, 2016 [24]	Iran	Vit D (600 IU/day)	No intervention	68/70	20	Birthweight
Roth, 2013 [25]	Bangladesh	Vit D3 (35,000 IU/week)	Placebo	80/80	12	Birthweight, birth length, head circumference
Sabet, 2012 [26]	Iran	Vit D3 (100000 IU/mol)	Placebo	25/25	12	Birthweight, birth length
Sablok, 2015 [27]	India	Vit D3 (single-intermitted dose depending upon the serum 25OHD levels)	No intervention	108/57	16	Birthweight, SGA
Yu, 2009 [28]	UK	Vit D2 (800 IU/d)	No intervention	60/59	13	LBW, SGA
		Vit D2 (single dose of 200000 IU)		60/59		

Abbreviations: Ca, Calcium; IU, International Unit; 25-OHD, 25-hydroxyvitamin D; LBW, low birth weight; SGA, small for gestational age; NA, not available.

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
