# Peer review of "Effects of Vitamin D Supplementation During Pregnancy on Birth Size: A Systematic Review and Meta-Analysis of Randomized Controlled Trials"

_nutrients, 2019, doi:10.3390/nu11020442_

Round 1
Reviewer 1 Report
This is a review paper of efficacy of vitamin D intake during pregnancy on pregnancy outcomes and infants. Maternal adequate vitamin D levels during pregnancy are required for regulating maternal immune system and skeletal growth of infant, therefore this is an important review. The review was well written; however, this review has some critical problems as below.
1. The Food and Nutrition Board at the Institute of Medicine of the National Academies suggested, a proper integration of vitamin D in pregnancy and in lactation is 15 micrograms (600 IU) per day; however, Vitamin D dosage is only 200IU in two studies (Asemi 2016 and Brough 2010). The authors should think of reanalysis without these two studies.
2. In this review, there is no statement of dosage dependent-effect of vitamin D. The review includes various dosage of vitamin D. The authors should analyze the correlation between dosage of vitamin D supplementation and odds ratio of events. And in discussion, the authors mentioned that vitamin D supplementation alone significantly increased birthweight, birth length and head circumference, but not in combination with other micronutrients. Is there a significantly difference of vitamin D dosage between vitamin D alone and combination studies? The authors need to discuss it.
3. In discussion, there is no statement of mechanisms for reducing the risk of LBW and SGA. Vitamin D has the effects on not only skeletal growth of infant, also regulation of maternal immune tolerance toward an infant. Therefore, maternal serum vitamin D insufficiency is associated with an increased risk of preterm birth (PMID 27213444 and PMID 28150405). The authors should discuss the mechanisms of the benefit of vitamin D supplementation on pregnancy and infants.
Author Response
Dear Editor,
thank you very much for considering our manuscript, and for your comments and those of independent Reviewers. We submit to your attention a revised version of the manuscript, in which we have considered all comments. The following List of change and answers to comments of Reviewers addresses all changes made in the manuscript using the track changes function in Microsoft Word.
Reviewer 1
This is a review paper of efficacy of vitamin D intake during pregnancy on pregnancy outcomes and infants. Maternal adequate vitamin D levels during pregnancy are required for regulating maternal immune system and skeletal growth of infant, therefore this is an important review. The review was well written; however, this review has some critical problems as below.
We are grateful to the Reviewer 1 for his/her positive comments.
1. The Food and Nutrition Board at the Institute of Medicine of the National Academies suggested, a proper integration of vitamin D in pregnancy and in lactation is 15 micrograms (600 IU) per day; however, Vitamin D dosage is only 200IU in two studies (Asemi 2016 and Brough 2010). The authors should think of reanalysis without these two studies.
As discussed in the Discussion section, we recognize that the Food and Nutrition Board recommends supplementation of 600 IU vitamin D per day. However, to the best of our knowledge, this recommendation is based on outcomes related to skeletal health. In contrast, evidence is currently inconclusive to drawn convincing assumptions on the usefulness of maternal vitamin D supplementation against LBW and SGA births. Thus, we appreciate the suggestion of Reviewer 1, but we think that excluding studies by Asemi and Brough could be inappropriate for our purpose.
2. In this review, there is no statement of dosage dependent-effect of vitamin D. The review includes various dosage of vitamin D. The authors should analyze the correlation between dosage of vitamin D supplementation and odds ratio of events. And in discussion, the authors mentioned that vitamin D supplementation alone significantly increased birthweight, birth length and head circumference, but not in combination with other micronutrients. Is there a significantly difference of vitamin D dosage between vitamin D alone and combination studies? The authors need to discuss it.
We agree with Reviewer 1 that evaluating the correlation of vitamin D dose with birth weight, birth length and head circumference could be useful to assess the dose-dependent effect. However, heterogeneity across studies (in terms of dose, duration and timing of supplementation) did not allow us to perform a meta-regression. With this in mind, in the Discussion section, we encouraged further studies evaluating the dose-dependent effect of vitamin D supplementation on birth outcomes. With respect to the second point, very few studies investigated the effect of vitamin D supplementation with other micronutrients, and hence we were not able to understand whether the combination with other supplements might affect the efficacy of intervention. Although single and combination interventions were based on similar vitamin D doses, we cannot exclude an antagonistic effect of other micronutrients. Given these limitations, we recommended further research to assess the dose-dependent effect of vitamin D alone or in combination with other micronutrients.
3. In discussion, there is no statement of mechanisms for reducing the risk of LBW and SGA. Vitamin D has the effects on not only skeletal growth of infant, also regulation of maternal immune tolerance toward an infant. Therefore, maternal serum vitamin D insufficiency is associated with an increased risk of preterm birth (PMID 27213444 and PMID 28150405). The authors should discuss the mechanisms of the benefit of vitamin D supplementation on pregnancy and infants.
According to Reviewer suggestion, we better discussed benefit of vitamin D supplementation on pregnancy and infants, including suggested references.
Reviewer 2 Report
This well written manuscript is a systematic review and meta-analysis of RCTs on the effects of vitamin D supplementation during pregnancy on birth size.
The methodology sounds fine; however, I have a few concerns:
1. Literature search: it is limited to 2 data bases and is not comprehensive. The authors should have searched the Cochrane Central Register of Controlled Trials at the minimum. They mentioned that unpublished studies were not included. Did authors made effort to contact the authors of those trials. This increase the risk of reporting and publication bias.
2. Statistical analysis: authors should explain why they decided to have P<0.1 for heterogeneity assessment.
3. Results: they included 13 RCTs in the meta-analysis but due to “eligible intervention group in 4 trials” they reported data on 17 group comparison. This approach has the potential of overestimating the positive effect of vitamin D supplementation on the outcome.
4. Table 1: needs to be reformatted as it is difficult to follow each study characteristic in the same row.
5. Line 203: 3.3.2 this needs to be changed to Birth Length.
6. Line 351- 360: Rate of confidence and supplementary file 5;
Birthweight: 1. there is risk of bias in the studies involved, 2. the 95% CI of the point estimates of the studies are wide and their sample size are small, there exist some imprecision in rating the confidence of the meta-analysis. The same principal applies to the rest of the outcomes.
I have not checked any original studies for double checking of the abstracted data, nor perform any literature review to rechecking included studies.
Author Response
Dear Editor,
thank you very much for considering our manuscript, and for your comments and those of independent Reviewers. We submit to your attention a revised version of the manuscript, in which we have considered all comments. The following List of change and answers to comments of Reviewers addresses all changes made in the manuscript using the track changes function in Microsoft Word.
Reviewer 2
This well written manuscript is a systematic review and meta-analysis of RCTs on the effects of vitamin D supplementation during pregnancy on birth size.
The methodology sounds fine; however, I have a few concerns:
We are grateful to Reviewer 2 for his/her positive comments.
1. Literature search: it is limited to 2 data bases and is not comprehensive. The authors should have searched the Cochrane Central Register of Controlled Trials at the minimum. They mentioned that unpublished studies were not included. Did authors made effort to contact the authors of those trials. This increase the risk of reporting and publication bias.
As suggested by reviewer 2, we now performed a literature search on the Cochrane Central Register of Controlled Trials database, using the same key words and filters (please see the new versions of Figure 1 and Supplementary Materials). However, no additional studies have been selected after removing duplicates.
We recognize that excluding unpublished studies could be a source of reporting and publication bias. However, as stated in the Cochrane Handbook for Systematic Reviews of Interventions, the inclusion of data from unpublished studies can itself introduce bias. Our reluctance to include grey literature is based on: i) the absence of peer-review of unpublished literature; ii) the studies that can be located may be an unrepresentative sample of all unpublished studies; iii) unpublished studies may be of lower methodological quality than published studies. To address publication biases, we visually assessed funnel plot asymmetry, followed by Begg’s test and Egger’s regression asymmetry test. As reported in the Results section, the funnel plot of the effect of vitamin D supplementation on birthweight appeared symmetric (Figure 8). Accordingly, Begg's rank correlation method and Egger's weighted regression method showed no sources of publication bias (p-values>0.05). With regard to other outcomes, due to the limited number of studies, the extent of publication biases cannot be excluded because the power of the tests for funnel plot asymmetry was too low to identify a real asymmetry.
2. Statistical analysis: authors should explain why they decided to have P<0.1 for heterogeneity assessment.
In our study, heterogeneity across studies was assessed according to Higgins and Thompson (doi:10.1002/sim.1186). Thus, we considered heterogeneity as significant if p<0.1 for Q-test based on the χ2 statistic and I² was greater than 30%.
3. Results: they included 13 RCTs in the meta-analysis but due to “eligible intervention group in 4 trials” they reported data on 17 group comparison. This approach has the potential of overestimating the positive effect of vitamin D supplementation on the outcome.
We recognize that including more group comparisons could have the potential of overestimating or underestimating the effect of vitamin D supplementation on the outcome. However, where possible, we performed subgroup analyses that classified group comparison into different subgroups. Anyway, we added it as a weakness of our study
4. Table 1: needs to be reformatted as it is difficult to follow each study characteristic in the same row.
We did our best to improve the formatting of Table 1 according to Journal guidelines.
5. Line 203: 3.3.2 this needs to be changed to Birth Length.
We are grateful for this comment, we changed title paragraph
6. Line 351- 360: Rate of confidence and supplementary file 5; Birthweight: 1. there is risk of bias in the studies involved, 2. the 95% CI of the point estimates of the studies are wide and their sample size are small, there exist some imprecision in rating the confidence of the meta-analysis. The same principal applies to the rest of the outcomes.
As suggested by Reviewer 2, we reviewed the Risk of Bias and Quality assessment and the Discussion sections, as well as supplementary file 5.
Reviewer 3 Report
Maugeri et al performed a systematic review and
meta-analysis of randomized controlled trials in order to assess the effect of
the supplementation of vitamin D during pregnancy on birth size. They selected
a total of 13 RCTs and confirmed the the well-established effect of maternal
vitamin D supplementation on weight at birth. The research topic is of
interest. The study is well designed and well-written. Statistical analysis is
appropriate. References are updated. The Authors recognize strenghts and
limitations of the study. They conclude in an honest way, suggesting benefits
in routine supplementation.
Author Response
Dear Editor,
thank you very much for considering our manuscript, and for your comments and those of independent Reviewers. We submit to your attention a revised version of the manuscript, in which we have considered all comments. The following List of change and answers to comments of Reviewers addresses all changes made in the manuscript using the track changes function in Microsoft Word.
Reviewer 3
Maugeri et al performed a systematic review and meta-analysis of randomized controlled trials in order to assess the effect of the supplementation of vitamin D during pregnancy on birth size. They selected a total of 13 RCTs and confirmed the the well-established effect of maternal vitamin D supplementation on weight at birth. The research topic is of interest. The study is well designed and well-written. Statistical analysis is appropriate. References are updated. The Authors recognize strenghts and limitations of the study. They conclude in an honest way, suggesting benefits in routine supplementation.
We are grateful to Reviewer 3 for his/her positive comments.
Round 2
Reviewer 1 Report
I am so disappointed with the author’s comments. They didn’t do anything for my suggestions. The author’s explanation doesn’t make sense. And I could not find the revised sentences which the author suggested.
Vitamin D has dosage dependent-effects; however, vitamin D dosage is only 200IU in two studies (Asemi 2016 and Brough 2010) as I mentioned. In previous study, the patients with vitamin D deficiency or insufficiency took 1000IU/d for 3 months of vitamin D supplementation, but half of patients could not achieve normal range of 25OHD (Ikemoto Y, et al. Nutrients. 2018. PMID: 30011861). 200IU is too small dose to analyze the effect of vitamin D.
And did the author check the correlation between dosage of vitamin D supplementation and odds ratio of events?
Hopefully the author will improve this paper.
Author Response
Dear Editor,
thank you very much for considering our manuscript, and for comments of independent Reviewers. We submit to your attention a revised version of the manuscript, in which we have considered all comments. The following List of change and answers to comments of Reviewer 1 addresses all changes made in the manuscript using the track changes function in Microsoft Word.
I am so disappointed with the author’s comments. They didn’t do anything for my suggestions. The author’s explanation doesn’t make sense. And I could not find the revised sentences which the author suggested.
We are sorry that Reviewer 1 was disappointed by our comments. In this new revised version of our manuscript we accomplished his/her suggestions.
Vitamin D has dosage dependent-effects; however, vitamin D dosage is only 200IU in two studies (Asemi 2016 and Brough 2010) as I mentioned. In previous study, the patients with vitamin D deficiency or insufficiency took 1000IU/d for 3 months of vitamin D supplementation, but half of patients could not achieve normal range of 25OHD (Ikemoto Y, et al. Nutrients. 2018. PMID: 30011861). 200IU is too small dose to analyze the effect of vitamin D.
We recognized that vitamin D supplementation of 200IU was too low. Thus, we performed a sensitivity analysis by the omission of studies with daily vitamin D dose of 200 IU (lines 142-143). However, the omission of these studies did not affect the association with birthweight (lines 202-203), , LBW (lines 256-258) and SGA (lines 271-273). The omission of Asemi 2016 and Brough 2010 was not possible for birth length and head circumference, as it resulted in only one study included in the meta-analysis.
And did the author check the correlation between dosage of vitamin D supplementation and odds ratio of events?
We recognized that vitamin D might have a dose-dependent effect on birth sizes. Thus, we performed a meta-regression on those outcomes with at least three studies evaluating daily vitamin D supplementation alone. Whereas, heterogeneity across studies - in terms of formulation, dose, duration and timing of supplementation - did not allow us to perform meta-regression of studies with single supplementation and/or combination of supplements (lines 145-151). However, meta-regression did not reveal a dose-dependent effect of vitamin D on birthweight (lines 204-205), birth length (lines 225-226), head circumference (lines 244-245) and SGA (lines 277-278). Meta-regression was not performed on LBW due to the limited number of studies. We also discussed these results in the Discussion section, suggesting that no significant dose-dependent effect of vitamin D could be attributed to the limited number of studies.